# TOWARDS STABLE AND EFFECTIVE REINFORCEMENT LEARNING FOR MIXTURE-OF-EXPERTS

## ABSTRACT

Recent advances in reinforcement learning (RL) have substantially improved the training of large-scale language models, leading to significant gains in generation quality and reasoning ability. However, most existing research focuses on dense models, while RL training for Mixture-of-Experts (MoE) architectures remains underexplored. To address the instability commonly observed in MoE training, we propose a novel router-aware approach to optimize importance sampling (IS) weights in off-policy RL. Specifically, we design a rescaling strategy guided by router logits, which effectively reduces gradient variance and mitigates training divergence. Experimental results demonstrate that our method significantly improves both the convergence stability and the final performance of MoE models, highlighting the potential of RL algorithmic innovations tailored to MoE architectures and providing a promising direction for efficient training of large-scale expert models.

## 1 INTRODUCTION

Reinforcement Learning (RL) has demonstrated strong potential in enhancing LLM reasoning through inference-time scaling, as exemplified by the OpenAI-o1 model (OpenAI, 2024). More recently, DeepSeek-R1 (Guo et al., 2025) has shown that reinforcement learning with verifiable rewards (RLVR), which relies on simple, rule-based reward functions, can elicit emergent reasoning abilities and deliver substantial performance gains on challenging tasks such as mathematical problem solving and program synthesis (Yang et al., 2025; Team et al., 2025a; Chen et al., 2025). In parallel, Mixture-of-Experts (MoE) architectures have emerged as an efficient approach to scaling model capacity (Fedus et al., 2022). By activating only a small subset of experts per token, MoE models achieve higher parameter efficiency while keeping computation cost nearly constant, making them particularly attractive for large-scale RL training where compute efficiency is critical.

Despite these advances, applying RLVR to MoE models remains highly challenging due to stability issues (Zheng et al., 2025; Chen et al., 2025; Yang et al., 2025). A central difficulty is *router fluctuation*: the set of experts selected for the same input token may vary significantly across policy updates (Dai et al., 2022; Zheng et al., 2025). Such routing drift not only increases the variance of importance sampling (IS) weights, but can also destabilize optimization and even cause reward collapse. Furthermore, most implementations adopt token-level IS ratios (Schulman et al., 2017), which are poorly aligned with the sequence-level rewards typically used in RLVR, introducing additional variance and further compounding instability. Prior work such as GSPO proposes computing IS ratios at the sequence level to mitigate this issue, yet it does not fundamentally address the instability introduced by router fluctuations.

We experimentally explored two natural stabilization strategies: freezing the router parameters and routing replay, where routing decisions are cached and reused across updates. However, both approaches proved unsatisfactory—freezing the router hampers model adaptability, while routing replay restricts exploration and degrades performance. These results suggest that rigid control over the router is suboptimal, and a more flexible mechanism is needed.

To address these challenges, we propose `Router-Shift Policy Optimization` (RSPO), an RL algorithm specifically designed for MoE architectures to achieve stable and efficient training. Instead of fully constraining the router, RSPO introduces a *router shift ratio*, computed from router scores between the current and old policies. This ratio quantifies the degree of routing deviation for

each token and is used to softly rescale IS weights. In doing so, `RSPO` reduces gradient variance and mitigates divergence caused by router instability, while preserving the router's capacity to adapt.

We validate the effectiveness of `RSPO` on the Qwen2.5 model for the countdown task and on Qwen3-30B-A3B across multiple mathematical reasoning benchmarks. Extensive experiments demonstrate that `RSPO` achieves more stable training and superior performance. Our main contributions are summarized as follows:

- We propose `RSPO`, which combines a router-aware rescaling strategy with sequence-level aggregation of importance sampling ratios, effectively stabilizing off-policy RL training for MoE models.
- Unlike router freezing or routing replay, `RSPO` adopts a soft adjustment mechanism: it leverages a router shift ratio to quantify routing deviation for each token and adaptively reweight updates, limiting overly large updates while retaining router flexibility.
- We evaluate `RSPO` on both the countdown task and multiple mathematical reasoning benchmarks, demonstrating its stability and effectiveness, and highlighting the importance of incorporating router-aware strategies and sequence-level importance weighting in RL for MoE models.

## 2 PRELIMINARIES

**Group Relative Policy Optimization (GRPO)**  Traditional reinforcement learning (RL) algorithms, such as Proximal Policy Optimization (PPO) (Schulman et al., 2017), have been widely applied to RL training of large language models (LLMs). However, PPO still suffers from high computational cost and challenges in tuning value model. To address these limitations, GRPO (Shao et al., 2024) builds upon PPO by removing the value model and introducing a group-relative advantage estimation.

Specifically, for a given query $x$, GRPO samples $G$ candidate responses $\{y_i\}_{i=1}^G$, computes their relative advantages within the group, and optimizes the following objective:

$$
\mathcal{J}_{\text{GRPO}}(\theta) = \mathbb{E}_{x \sim \mathcal{D},\, \{y_i\}_{i=1}^G \sim \pi_{\theta_{\text{old}}}(\cdot|x)} \left[ \frac{1}{G} \sum_{i=1}^G \frac{1}{|y_i|} \sum_{t=1}^{|y_i|} \min\Big( w_{i,t}(\theta)\, \hat{A}_{i,t},\, \text{clip}\big(w_{i,t}(\theta), 1-\epsilon, 1+\epsilon\big) \hat{A}_{i,t} \Big) \right],
$$

$$(1)$$

where $G$ denotes the number of responses generated for each query $x$. For each token $y_{i,t}$, the importance sampling ratio $w_{i,t}(\theta)$ and the group-normalized advantage $\hat{A}_{i,t}$ are given by:

$$
w_{i,t}(\theta) = \frac{\pi_\theta(y_{i,t} \mid x, y_{i,<t})}{\pi_{\theta_{\text{old}}}(y_{i,t} \mid x, y_{i,<t})}, \qquad \hat{A}_i = \hat{A}_{i,t} = \frac{r(x,y_i) - \text{mean}\big(\{r(x,y_i)\}_{i=1}^G\big)}{\text{std}\big(\{r(x,y_i)\}_{i=1}^G\big)}. \tag{2}
$$

**Group Sequence Policy Optimization (GSPO).**  Unlike GRPO, which performs importance sampling and clipping at the token level, Group Sequence Policy Optimization (GSPO) (Zheng et al., 2025) defines the importance sampling ratio at the *sequence level* and applies clipping accordingly. This modification corrects the misalignment between sequence-level rewards and token-level importance sampling ratios present in GRPO under the RLVR objective. The GSPO optimization objective can be formulated as:

$$
\mathcal{J}_{\text{GSPO}}(\theta) = \mathbb{E}_{x \sim \mathcal{D},\, \{y_i\}_{i=1}^G \sim \pi_{\theta_{\text{old}}}(\cdot|x)} \left[ \frac{1}{G} \sum_{i=1}^G \min\Big( s_i(\theta)\widehat{A}_i,\, \text{clip}\big(s_i(\theta), 1-\varepsilon, 1+\varepsilon\big) \widehat{A}_i \Big) \right],
$$

where the sequence-level advantage $\hat{A}_i$ usually comes from a rule based reward and calculate in the same way as GRPO:

$$
\hat{A}_i = \frac{r(x,y_i) - \text{mean}\big(\{r(x,y_j)\}_{j=1}^G\big)}{\text{std}\big(\{r(x,y_j)\}_{j=1}^G\big)}, \tag{3}
$$

The sequence-level importance ratio $s_i(\theta)$ based on sequence likelihood is defined as:

$$s_i(\theta) = \left( \frac{\pi_\theta(y_i \mid x)}{\pi_{\theta_{\text{old}}}(y_i \mid x)} \right)^{\frac{1}{|y_i|}} = \exp\left( \frac{1}{|y_i|} \sum_{t=1}^{|y_i|} \log \frac{\pi_\theta(y_{i,t} \mid x, y_{i,<t})}{\pi_{\theta_{\text{old}}}(y_{i,t} \mid x, y_{i,<t})} \right). \tag{4}$$

Essentially, this replaces the arithmetic mean of the importance sampling ratios across the sequence with their geometric mean.

**Geometric-Mean Policy Optimization (GMPO).** While GSPO operates at the sequence level by defining a sequence-level importance sampling ratio and applying clipping over entire sequences, GMPO (Zhao et al., 2025) retains the token-level decomposition but replaces the arithmetic mean aggregation of per-token importance ratios with their geometric mean. In GMPO, each token's ratio (new/old) is treated multiplicatively, and the $|y|$-th root (or equivalently the sum of log ratios) is used to obtain a single sequence-level ratio, which is more robust against extreme individual token ratios. Unlike GSPO, where extreme token-level ratios may still dominate the overall sequence likelihood, GMPO reduces the influence of such outliers while preserving token-level granularity.

## 3 METHOD

### 3.1 MOTIVATION

Modern Mixture-of-Experts (MoE) models often have extremely large model sizes (Guo et al., 2025; Liu et al., 2024; Yang et al., 2025; Team et al., 2025a),, making off-policy reinforcement learning (RL) crucial for maintaining training efficiency on reasoning tasks. However, as highlighted by GSPO, two key issues arise when applying traditional GRPO to MoE models:

(1) **Routing fluctuations**: As shown in Figure 1 after policy updates, the set of experts activated for the same token may change (approximately 10% of experts differ according to prior studies).And even when the selected experts remain unchanged, their routing probabilities may still shift.Such changes cause substantial fluctuations in the importance-sampling ratios, frequently triggering the clipping mechanism and introducing additional variance into the training process.This phenomenon was also reported previously in STABLEMOE (Dai et al., 2022);

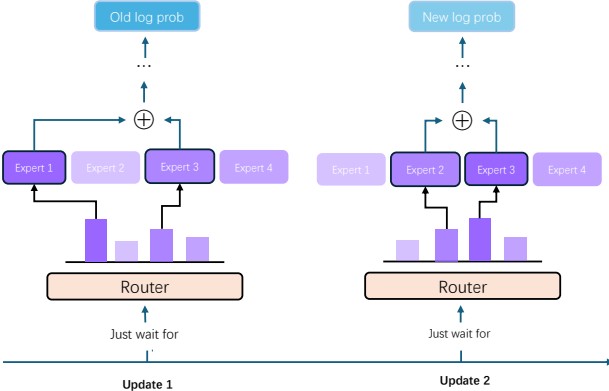

Figure 1: Router fluctuation in off-policy training.

(2) **Variance mismatch**: Most implementations of GRPO align sequence-level advantages using token-level importance sampling ratios, creating a mismatch: token-level variance does not accurately reflect sequence-level variance. In MoE settings, this effect is amplified by multiple experts per token, often triggering clipping and destabilizing training. Combined with routing fluctuations, this variance mismatch can significantly hinder stable and efficient learning.

A straightforward mitigation is routing replay (Zheng et al., 2025), which reuses the expert assignments from the old policy when computing current log-probabilities, thus eliminating routing drift.

However, this approach constrains router updates and incurs significant memory and communication overhead, harming scalability and performance. GSPO handle the variance mismatch by adopting sequence-level importance ratios and clipping, which better aligns the optimization objective with rule based rewards and it is functionally similar with GMPO (using a geometric rather than arithmetic mean). Nonetheless, GSPO does not fundamentally resolve routing distribution drift, and its sequence-level clipping can over-prune tokens, potentially discarding useful gradient information.

## 3.2 OUR APPROACH

To address these limitations, we propose `Router-Shift Policy Optimization` (RSPO), a routing-aware off-policy RL algorithm for MoE models. RSPO retains the use of sequence-level importance ratios but applies token-level clipping to reduce information loss. Furthermore, we introduce a router shift ratio that measures the deviation of router distributions between the current and old policies. This ratio is used both to reweight token-level importance ratios and to softly clip tokens exhibiting severe routing drift, thereby stabilizing training without freezing the routert. The optimization objective of RSPO is formulated as:

$$\mathcal{J}_{\text{RSPO}}(\pi_\theta) = \mathbb{E}_{x \sim \mathcal{Q}, \{o_i\}_{i=1}^G \sim \pi_{\theta_{\text{old}}}(\cdot|x)}$$

$$\frac{1}{G} \sum_{i=1}^G \left\{ \prod_{t=1}^{|o_i|} \left( \min\left[ (w_{i,t}(\theta))^{\text{sgn}(\hat{A}_i)}, \text{clip}\big((w_{i,t}(\theta))^{\text{sgn}(\hat{A}_i)}, \epsilon_1, \epsilon_2\big) \right] \cdot \gamma_{i,t} \right)^{\text{sgn}(\hat{A}_i)} \right\}^{\frac{1}{|o_i|}} \hat{A}_i \tag{5}$$

The terms $w_{i,t}(\theta)$ and $\hat{A}_i$ are defined in the same way as in GRPO, $sgn(\hat{A}_i)$ represents the sign of the advantage, guaranteeing that the clipping mechanism functions as intended. The key innovation of our method is the introduction of router shift ratio, which quantifies the degree of routing drift and dynamically adjusts the magnitude of policy updates. Specifically, $\gamma_{i,t}$ denotes the router shift ratio, and $r_\phi^{(\ell)}$ represents the routing score of expert $e_{i,t}^{(\ell)}$ at layer $\ell$. We compute the deviation between $\phi_{\text{old}}$ and $\phi$ by averaging the scores of the top-$K$ experts that were activated under the old policy during the log-probability computation. Since routing deviations may accumulate across layers, we aggregate the per-layer router shifts multiplicatively. The resulting $\gamma_{i,t}$ is applied as a reweighting factor after clipping at the token level, thus preserving the original clipping behavior while further down-weighting tokens that experience excessive routing drift. This design improves the stability of the training without interfering with the variance control mechanism provided by clipping.

$$\gamma_{i,t} = \exp\left( -\frac{1}{L} \sum_{\ell=1}^L \frac{1}{K} \sum_{k=1}^K \left| \log r_\phi^{(\ell)}\big(e_{i,t}^{(\ell,k)} \mid x, y_{i,<t}\big) - \log r_{\phi_{\text{old}}}^{(\ell)}\big(e_{i,t}^{(\ell,k)} \mid x, y_{i,<t}\big) \right| \right), \tag{6}$$

$$\gamma_{i,t} = \max\big(\gamma_{i,t}, \gamma_{\min}\big).$$

This construction ensures that $\gamma_{i,t} \in (0, 1]$ reflects the degree of router distribution drift, allowing tokens with larger routing shifts to be down-weighted or clipped during optimization.

Another important aspect of our method is the use of a geometric mean for aggregating importance sampling ratios at the sequence level. As discussed earlier, the variance of off-policy updates can be further amplified in MoE architectures due to routing drift. Our experiments show that applying a geometric mean to the importance ratios significantly improves training stability, whether clipping is performed at the token level (GMPO) or at the sequence level (GSPO). As shown in Figure 2, the model trained with GRPO experiences reward collapse on the small-scale setting, while GSPO and GMPO remain stable. On the large-scale Qwen3-30B-A3B model, GRPO achieves substantially lower reward compared to GSPO and GMPO, further confirming the advantage of using geometric mean aggregation.

Our proposed Router-Shift Policy Optimization (RSPO) method can be seamlessly integrated into existing reinforcement learning training pipelines. Specifically, it adjusts the importance sampling

ratios based on the token-level router shift, mitigating instability caused by routing fluctuations during training.:

**Router-Shift Policy Optimization (PyTorch Implementation)**

```
# ----------- Step 1: Compute router shift ratio -----------
# Measure the routing probabilities change between steps
old_log_probs = torch.log(torch.clamp(old_topk_probs, min=eps))
current_log_probs = torch.log(torch.clamp(selected_current_probs, min=
    eps))
delta_log_probs = torch.abs(current_log_probs - old_log_probs)
router_shift_ratio = torch.exp(-delta_log_probs).mean(dim=-1)

# ----------- Step 2: Clip router shift ratio -----------
# Avoid extreme values that may destabilize training
router_clip_mask = (router_shift_ratio < clip_threshold) & (
    response_mask > 0)

# ----------- Step 3: Apply router shift ratio -----------
# Smooth the importance sampling ratio
log_router_weights = torch.log(torch.clamp(router_shift_ratio, min=1e
    -8))
negative_approx_kl_min = negative_approx_kl_min + log_router_weights
ratio = torch.exp(negative_approx_kl_min)
```

## 4 EXPERIMENTS

### 4.1 EXPERIMENTAL DETAILS

**Training Setup.** We first conduct exploratory experiments and ablation studies on a small-scale MoE model built on the Qwen2.5 (Qwen et al., 2025) architecture to validate the effectiveness of our proposed approach and better understand its behavior. We pretrain the model on the Countdown task, with a total parameter size of 385M and 85M activated parameters. Each layer contains 8 experts, with top-$k = 1$ expert routing. The pretraining is conducted with a learning rate of $1 \times 10^{-4}$ on 8 H100 GPUs. The RL implementation is built upon the verl training library. For large-scale evaluation, we compare our method against GRPO (Shao et al., 2024), GSPO (Zheng et al., 2025), and GMPO (Zhao et al., 2025) on the Qwen3-MoE-30B-A3B (Qwen et al., 2025) model. All experiments are conducted under an off-policy training setting with a response length of $8$k, a global batch size of $128$, and a mini-batch size of $64$. For GRPO, we adopt the commonly used clipping range of $0.2$, while for GSPO and GMPO we follow the recommended hyperparameter settings reported in their respective papers.

**Training Datasets.** For training, we use the DeepScaleR (Luo et al., 2025) dataset, which contains approximately 40,000 unique mathematics problem–solution pairs. This dataset is compiled from multiple sources, including: AIME (American Invitational Mathematics Examination) problems from 1984–2023, AMC (American Mathematics Competition) problems prior to 2023, the Omni-MATH (Gao et al., 2024) dataset, and the Still dataset. During RL training, we employ a rule-based reward function that evaluates the correctness of generated answers against the reference solutions.

**Evaluation Datasets.** We assess the effectiveness of our approach on five mathematical reasoning datasets spanning a wide range of difficulty levels, following the experimental protocol of Dr.GRPO. The evaluation includes: AIME24, a set of 30 high-school level problems from the 2024 American Invitational Mathematics Examination; AMC, with 83 moderately challenging multiple-choice questions; MATH500, a 500-problem subset of the MATH (Hendrycks et al., 2021) dataset covering algebra, geometry, and number theory; Minerva (MIN) (Lewkowycz et al., 2022), consisting of 272 graduate-level problems requiring multi-step reasoning; and OlympiadBench (Huang et al., 2024), which contains 675 advanced olympiad-style questions. Together, these benchmarks comprehensively evaluate reasoning capabilities across problem domains and difficulty scales. We report results using the Pass@1 metric, which measures whether a single generated solution is correct.

Table 1: Performance comparison on five mathematical reasoning benchmarks.

| Method | AIME24 | AMC | MATH500 | Minerva | OlympiadBench | Avg. |
|---|---|---|---|---|---|---|
| Base | 80.4 | 90.0 | 90.7 | 47.7 | 62.0 | 74.2 |
| GRPO (Shao et al., 2024) | 77.0 | 82.5 | 91.8 | 48.2 | 58.1 | 71.5 |
| GSPO (Zheng et al., 2025) | 80.4 | 95.0 | 93.6 | 48.9 | 64.0 | 76.4 |
| GMPO (Zhao et al., 2025) | 80.1 | 92.5 | 94.2 | 49.3 | 65.9 | 76.4 |
| RSPO | 80.1 | 95.0 | 94.2 | 50.7 | 65.8 | **77.1** |

For AIME24, we report the mean accuracy over 32 runs. For all benchmarks, we use deterministic decoding by setting the temperature to $0.0$ and produce one answer per problem.

## 4.2 MAIN RESULTS

Table 1 provides a comprehensive comparison of our proposed RSPO method against several established baselines, including GRPO, GSPO, and GMPO, across five widely used mathematical reasoning benchmarks: AIME24, AMC, MATH500, Minerva, and OlympiadBench. The evaluation was conducted using the Qwen3-30B-A3B model. From the results, it is evident that RSPO consistently outperforms the competing methods across nearly all benchmarks, with particularly notable gains in Minerva and OlympiadBench. While GSPO and GMPO already show strong performance on MATH500 and OlympiadBench, RSPO achieves the highest average score of 77.1, highlighting its effectiveness in enhancing the model's reasoning capabilities. These results underscore the robustness and generalizability of RSPO for complex mathematical reasoning tasks.

Furthermore, Figure 2 illustrates the reward progression and validation score trends during training for RSPO and the baseline methods. Notably, GRPO exhibits a pronounced performance collapse around 200–500 training steps, particularly evident in the sharp decline in validation score, indicating severe instability during early training. Although GRPO eventually recovers partially, its performance remains highly volatile and consistently lower than the other methods throughout training. GSPO and GMPO show more stable trajectories, but RSPO consistently achieves the highest and most stable

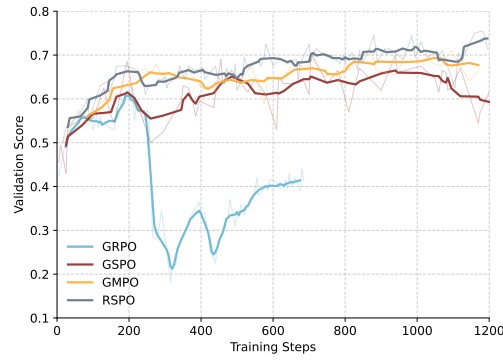

Figure 2: Compare RSPO with other baselines

validation scores across all training steps, demonstrating both rapid convergence and robustness. This behavior highlights RSPO's ability to stabilize training dynamics in multi-expert settings, whereas GRPO not only lags behind other baselines in average performance but, as shown in Table 1, even underperforms the base model in terms of mean validation score.

The reward and test score trajectories further support the conclusion that GRPO suffers from instability when applied to MoE models. In contrast, RSPO not only achieves consistently better results across benchmarks but also demonstrates significantly more stable training behavior. These findings highlight the critical role of incorporating the router shift ratio, which contributes both to improved performance and training robustness in MoE-based RL fine-tuning.

## 4.3 ANALYSIS

**Additional Attempts on Router Stabilization** In addition to our main approach, we also investigated several alternative strategies aimed at improving the stability of MoE RL training.

One straightforward idea is to freeze the router during training, with the expectation that eliminating router updates could mitigate the fluctuations observed in expert selection. While simple to imple-

ment, this approach assumes that the initial router configuration is already well aligned with the optimization objective, which may not necessarily hold in practice.

Beyond freezing, we drew inspiration from the router replay technique proposed in GSPO, which provides a more fundamental direction for addressing router drift. To this end, we designed and tested two distinct variants of router replay.

**(1) Copying logits from the old policy.** In this variant, the router logits of the old policy are directly copied to replace those of the current policy. As a result, both expert selection and expert weighting become fully aligned with the old policy. However, since the router logits are no longer computed by the current model, the router in the current policy cannot propagate gradients. This restriction significantly limits the router's ability to update and adapt to the evolving training dynamics.

**(2) Reusing expert indices.** Alternatively, instead of transferring logits directly, we record the indices of the experts activated by the old policy and enforce the current policy to select experts according to these stored indices. This strategy preserves the discrete expert choices of the old policy while still allowing the current router to compute its own logits, albeit with constrained selection.

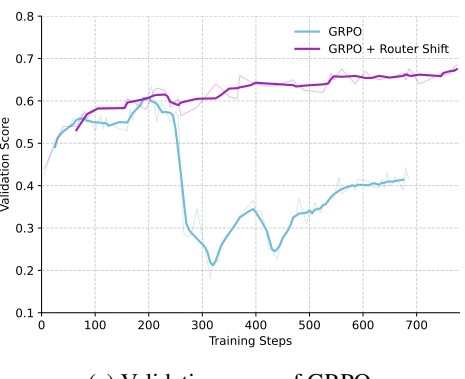

Figure 3: Routing replay and freeze router

As illustrated in Figure 3, we conducted experiments on Qwen2.5 to systematically evaluate these three methods: freezing the router, copying old router logits, and reusing expert indices. The empirical results indicate that none of these approaches led to satisfactory improvements in training stability. These findings suggest that while intuitive, such heuristic methods are insufficient to address the inherent instability of MoE routers, further highlighting the necessity of more principled solutions such as the proposed RSPO.

**Combine Router Shift with Other Algorithms** An appealing property of our approach is its inherent compatibility with a wide range of reinforcement learning algorithms, making it a flexible component that can be readily integrated into different training paradigms. To empirically validate this property, we applied the proposed router shift ratio in combination with two representative baselines, GSPO and GRPO. Specifically, we incorporated router shift into the training and evaluation of the Countdown task using the Qwen2.5 MoE model, as illustrated in Figure 4.

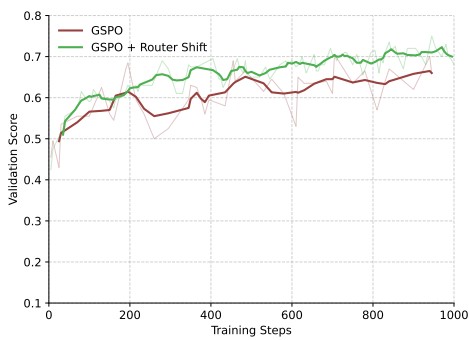

(a) Validation score of GRPO  (b) Validation score of GSPO

Figure 4: Combine router shift with other algorithm

The results clearly show that introducing router shift consistently improves both training stability and final task performance when paired with either GSPO or GRPO. For GSPO, the integration of router shift further smooths the optimization trajectory, reducing fluctuations that commonly arise from the stochasticity of expert selection. For GRPO, which is more prone to instability in MoE settings, router shift effectively alleviates divergence issues and enables more reliable convergence. In both cases, the observed improvements highlight that router shift does not interfere with the core mechanisms of the underlying algorithms but instead acts as a stabilizing and performance-enhancing augmentation.

Taken together, these findings underscore the versatility of our technique and demonstrate its potential to be seamlessly combined with existing RL methods. By enhancing stability while boosting performance, router shift offers a general solution for improving the robustness of reinforcement learning on MoE-based large language models, beyond the scope of a single algorithmic framework.

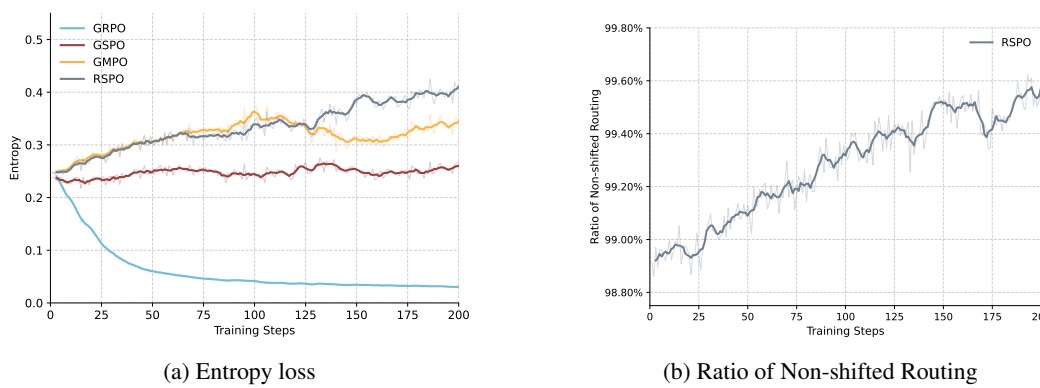

(a) Entropy loss           (b) Ratio of Non-shifted Routing

Figure 5: Training stability signals of Qwen3-30B-A3B

**Training Stability**     To better understand the effect of RSPO on training dynamics, we monitor several key stability-related metrics throughout the training of Qwen3-MoE-30B-A3B, including token-level entropy and the ratio of non-shifted routing.

As illustrated in Figure 5, RSPO consistently maintains higher token entropy compared to baseline methods, suggesting that it preserves a more diverse distribution over experts and mitigates premature collapse into suboptimal routing patterns.

The ratio of non-shifted routing offers valuable insights into the model's training dynamics. This metric reflects the proportion of tokens for which the current policy model's routing decisions match those of the previous policy model, i.e., no routing shift occurs. The steadily increasing trend of this ratio during training shows that, as the model converges, the routing policy becomes increasingly consistent with its previous version, reducing the need for shift corrections. Early in training, a larger fraction of tokens requires routing shifts to maintain stability, but as optimization progresses, the policy naturally aligns with prior decisions, demonstrating that RSPO's reweighting mechanism effectively promotes stable and consistent routing over time.

## 5   RELATED WORK

### 5.1   REINFORCEMENT LEARNING FOR LLM

Recently, the emergence of DeepSeek R1 (Guo et al., 2025) has demonstrated the significant potential of combining reinforcement learning (RL) with reasoning for pushing the performance boundaries of large language models (LLMs). At the core of R1 lies the Group Relative Policy Optimization (GRPO) (Shao et al., 2024) algorithm, which represents an improvement over the well-known Proximal Policy Optimization (PPO) (Schulman et al., 2017) algorithm. GRPO estimates advantages within groups, thereby eliminating the need for an expensive value function model while maintaining performance comparable to PPO. The success of R1 has sparked widespread interest in GRPO

and inspired the development of numerous variants. For instance, DAPO (Yu et al., 2025) introduces techniques such as dynamic sampling and higher clipping thresholds, addressing challenges related to training efficiency and stability. Dr. GRPO (Liu et al., 2025) focuses on mitigating length bias by removing the length and standard deviation normalization terms in GRPO, thereby reducing optimization bias and improving token efficiency.More recently, several studies have highlighted issues with the token-level importance sampling ratio used in GRPO, which can lead to increased variance. To address this, GMPO (Zhao et al., 2025) proposes maximizing token-level rewards using a geometric mean, resulting in more stable training dynamics. Similarly, GSPO (Zheng et al., 2025) approaches the problem from the sequence-level importance ratio perspective, ultimately also converging on a geometric mean formulation for enhanced stability. Notably, GSPO reports that this geometric mean approach is particularly effective for reinforcement learning training in Mixture-of-Experts (MoE) models.

## 5.2 STABILITY IN MoE TRAINING.

Mixture-of-Experts (MoE) models have emerged as a key technique for scaling neural networks to trillions of parameters while maintaining computational efficiency by sparsely activating only a small subset of experts per token. However, this sparse activation introduces unique challenges, including expert under-utilization, load imbalance, and routing instability. Severe load imbalance can lead to some experts being overloaded while others receive few or no tokens, resulting in inefficient use of model capacity and degraded convergence. Switch Transformer (Fedus et al., 2022) addresses these challenges by introducing an auxiliary load-balancing loss to encourage uniform expert utilization and a capacity factor to cap the number of tokens routed to each expert, thus preventing overload. While effective, large auxiliary losses can introduce non-negligible gradient interference with the main training objective. Wang et al. mitigate this by proposing the Loss-Free Balancing method, which dynamically adjusts expert-wise biases on routing scores before top-$k$ selection to balance expert loads without introducing additional loss terms, thereby avoiding gradient interference and improving the attainable model performance. StableMoE (Dai et al., 2022) further identifies routing fluctuation as a key source of instability, proposing to distill a stable teacher router and freeze it during training to reduce token assignment variance. Another line of work focuses on improving gradient flow through non-differentiable top-$k$ routing by using differentiable relaxations such as Gumbel-Softmax or straight-through estimators (Wang et al., 2024b; Puigcerver et al., 2023; Zhou et al., 2022), thereby reducing gradient variance and enabling end-to-end optimization.

More recently, researchers have observed that MoE models are particularly unstable under reinforcement learning (RL) training, where reward sparsity and high-variance policy gradients exacerbate routing fluctuations. To address this, several approaches aim to stabilize MoE routers during RL fine-tuning. For instance, GSPO (Zheng et al., 2025) stabilizes off-policy updates by reusing expert assignments from previous policies and clipping sequence-level importance sampling ratios, effectively reducing update variance. Ring-lite (Team et al., 2025b) introduces constrained token-level routing budgets to regularize expert selection and further reduce variance. Despite these advances, understanding the interplay between routing dynamics, gradient variance, and RL credit assignment remains an open research direction, motivating methods like RSPO that explicitly account for router shift when shaping policy updates.

## 6 CONCLUSION

In this work, we propose RSPO, a method that stabilizes policy training and improves final performance by employing sequence-level importance sampling for more accurate gradient estimation, and introducing a router shift ratio to down-weight tokens that exhibit excessive router shift during off-policy training. This approach also helps maintain token entropy at a relatively high level.Extensive experiments on mathematical reasoning benchmarks demonstrate that RSPO consistently outperforms current baselines in both stability and overall effectiveness.This work represents a step forward in developing more stable and effective RL training methods tailored for mixture-of-experts (MoE) models, facilitating their reliable deployment in large-scale reasoning tasks.

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

# A   APPENDIX

## REPRODUCIBILITY STATEMENT

We emphasize that our experimental setup is designed to be highly reproducible. All model architectures, training hyperparameters, and evaluation protocols are described in detail in the paper. The training datasets we use are publicly available, and our implementation is based on open-source deep learning frameworks. These choices ensure that researchers can reliably reproduce our results and build upon our work.

## ETHICS STATEMENT

This work focuses on improving the efficiency and stability of Mixture-of-Experts models. It does not involve the collection or use of sensitive personal data. Potential societal impacts include both positive contributions by enabling more efficient large-scale models and risks such as misuse for generating harmful content, which we mitigate by recommending alignment and safety checks before deployment.

## LLM USAGE STATEMENT

We acknowledge the use of a large language model (OpenAI ChatGPT) as a general-purpose assistive tool during the preparation of this manuscript. Specifically, the LLM was employed for **language polishing and improving readability** of the text (e.g., refining grammar, rephrasing sentences for clarity, and adjusting tone to match academic style). No part of the research design, data collection, analysis, or substantive interpretation of results was performed by the LLM. The authors take full responsibility for the accuracy and integrity of all contents presented in this paper.

