# OpenReview forum: "Towards Stable and Effective Reinforcement learning for Mixture-of-Experts"
_ICLR.cc/2026/Conference — ICLR 2026 Conference Withdrawn Submission_

### Official Review · Reviewer_mvEe · 2025-10-28

**Soundness:** 3
**Presentation:** 3
**Contribution:** 2
**Rating:** 4
**Confidence:** 3

**Summary:**

the paper presents a novel approach to optimize importance sampling of weights for off-polich RL specifically for RL training for Mixture-of-Experts (MoE) architectures that still suffers from instability. the authors present RSPO as an algorithm for this problem.  The core novelty of RSPO is the router shift ratio. Unlike prior attempts to stabilize MoE training by rigidly freezing the router or replaying old routing decisions, RSPO uses this ratio as a soft adjustment mechanism. It dynamically adapts the magnitude of policy updates based on the degree of routing drift for each token, maintaining router flexibility while preventing instability. The authors present experimental results on mathematical reasoning tasks.

**Strengths:**

The method specifically addresses unique MoE failure modes. RSPO demonstrates significantly more stable training than GRPO (reward collaps) and does outperform GSPO and GMPO on the evaluation tasks but only marginally. They provide some analysis on why the RSPO prevents collaps.

**Weaknesses:**

I think the results are not super strong, especially comparing with GSPO and GMPO does not really show any significant benefit at least on the tasks that have been presented. Another weakness is that the evaluation tasks are all from within a very similar domain, raising questions about how well this method would work on a broader set of domains.

**Questions:**

Have you done experiments on different tasks?

---

### Official Review · Reviewer_NVzX · 2025-10-30

**Soundness:** 3
**Presentation:** 2
**Contribution:** 2
**Rating:** 2
**Confidence:** 3

**Summary:**

The paper studies instability in reinforcement learning for Mixture-of-Experts (MoE) language models. The authors identify routing fluctuation, the changes in which experts are selected across policy updates, as a major source of variance in importance-sampling ratios.
To address this, they propose Router-Shift Policy Optimization (RSPO), which adds a router-shift coefficient gamma(i,t) to down-weight tokens that experience large changes in routing logits between the old and current policies. They also switch from sequence-level to token-level clipping to further stabilize training. Empirical results on Qwen2.5 and Qwen3-30B-A3B show that RSPO improves training stability and final accuracy across math-reasoning benchmarks compared to GRPO, GSPO, and GMPO.

**Strengths:**

- The paper pinpoints a practical instability in MoE RL training.
- The proposed fix is simple but sensible. The router-shift weighting makes intuitive sense and is easy to integrate.
- The topic is timely and practically significant for large-scale RLHF/RLVR pipelines.
- Experiments demonstrate performance boosts and stability improvements on competitive models. The ablations on router freezing and replay variants add useful context.

**Weaknesses:**

- The novelty over GSPO/GMPO is limited. The router-shift term is the main difference, and there’s no theoretical analysis to back its variance-reduction claim.
- Important details (values for K, gamma_min, and clipping thresholds) are missing, making it hard to judge reproducibility or sensitivity.
- Lack of quantitative understanding of the stability improvement. The paper does not measure or visualize the actual variance reduction, distribution of importance ratios, or clipping rates. Including such diagnostics would clarify the mechanism behind the reported stability gains.
- The experiments focus only on math reasoning; it’s unclear whether the method generalizes to other domains.
- There’s no deeper analysis of why RSPO stabilizes training, for example, variance or clipping-rate plots would help.
- Minor typos and formatting issues remain.
- Some figure captions are not self-contained and require reading the main text for interpretation.

**Questions:**

- Can you provide empirical or theoretical evidence that gamma(i,t) reduces gradient variance compared to GSPO/GMPO?

- What exact hyperparameters were used (K, L, gamma_min, epsilon1, epsilon2), and how sensitive are results to them?

- What’s the compute/memory overhead of storing router states?

- Why mix geometric aggregation with token-level clipping? Would sequence-level clipping behave differently?

- Were all baselines tuned under the same compute budget? GRPO seems under-optimized.

- Do the gains hold in other RLVR setups, e.g., code synthesis or denser reward tasks?

- Could you add diagnostics (IS-ratio distributions, clipping frequency, gradient-norm variance) to back the stability claim?

- How were gradients handled in the router-replay variants?

- How many random seeds were used in each experiment, and how consistent are results across seeds?

**Details Of Ethics Concerns:**

There are no ethics concerns to report.

---

### Official Review · Reviewer_pQzx · 2025-10-31

**Soundness:** 3
**Presentation:** 2
**Contribution:** 3
**Rating:** 4
**Confidence:** 3

**Summary:**

This paper addresses  the instability of Reinforcement Learning (RL) training when applied to Mixture-of-Experts (MoE) architectures in scaling large language models., particularly the so-called "router fluctuation"—where the set of activated experts for a given token changes between policy updates—as a primary source of high variance and training divergence in off-policy RL like GRPO.  The authors proposes a  new algorithm, Router-Shift Policy Optimization (RSPO), which introduces a "router shift ratio" to quantify and penalize excessive routing deviations. Extensive experiments on various mathematical reasoning benchmarks show some promising applications.

**Strengths:**

(1) The paper is well-motivated. It deals with a challenging problem and proposes a well-explored solution.
(2)The paper performss a Comprehensive Empirical Validation: The evaluation is rigorous, using both small-scale ablation studies and large-scale models across five diverse mathematical reasoning benchmarks.
(3) The algorithm design is well-elaborated.

**Weaknesses:**

(1) Limited Task and Model Scope:  The paper's results are limited to mathematical reasoning tasks and Qwen family of models.
(2) Ablation Study Depth: While the main components are justified, a more detailed ablation study within RSPO itself—for instance, isolating the individual contribution of the router shift ratio from the geometric mean aggregation—would provide deeper insight into which aspect is most critical for the observed gains.

**Questions:**

(1) Generalization: How do you anticipate RSPO would perform on non-reasoning tasks or MoE architectures with fundamentally different routing mechanisms (e.g., Expert Choice)? Do you think the router shift ratio is universally applicable?
(2) Reward Function Sensitivity: The experiments use a rule-based reward for mathematical correctness. How sensitive is RSPO's stability to the nature of the reward signal, particularly with denser or sparser rewards?
(3)  Cost: Could you elaborate on the observed computational overhead of RSPO?

---

### Official Review · Reviewer_P5EK · 2025-10-31

**Soundness:** 3
**Presentation:** 3
**Contribution:** 3
**Rating:** 6
**Confidence:** 3

**Summary:**

This paper tackles the training instability of applying reinforcement learning to Mixture-of-Experts (MoE) models, a problem caused by "router fluctuation"—inconsistent expert selection for the same token across policy updates. This fluctuation creates high variance in importance sampling weights, often leading to training instabilities. The authors propose an algorithm that introduces a "router shift ratio" to quantify this routing deviation on a per-token basis. This ratio is then used to softly down-weight the importance sampling weights of tokens with significant drift. Experimental results on a suite of benchmarks, especially mathematical reasoning, show that this "soft adjustment" mechanism leads to more stable training and improved final performance over standard RL baselines.

**Strengths:**

The proposed method addresses a contemporary stability problem at the heart of the LLM post training pipeline.
The paper presents convincing empirical evidence that the proposed methods work on modern open weight MoE models such as Qwen3-30B-A3B and on a range of contemporary RL benchmark tasks.
Last but not least, while maybe a bit ad-hoc, the proposed solution is relatively simple and boils down to a soft-regularization term in contrast to some recent alternative methods that instead propose hard constraints.

**Weaknesses:**

While the paper presents end-to-end results for the proposed methods, there are only few detailed analysis and ablation studies, even though the design includes several choices such as using the absolute log-difference or aggregating them multiplicatively. While these choices seem intuitively reasonable, they are often neither theoretically or experimentally confirmed.

**Questions:**

Pretraining often involves auxiliary load-balancing losses. How do these interact with the proposed method?

---

### Note · Authors · 2025-12-02

I have read and agree with the venue's withdrawal policy on behalf of myself and my co-authors.